# Pilot-RCT Finds No Evidence for Modulation of Neuronal Networks of Auditory Hallucinations by Transcranial Direct Current Stimulation

**DOI:** 10.3390/brainsci12101382

**Published:** 2022-10-12

**Authors:** Lynn Marquardt, Alexander R. Craven, Kenneth Hugdahl, Erik Johnsen, Rune Andreas Kroken, Isabella Kusztrits, Karsten Specht, Anne Synnøve Thomassen, Sarah Weber, Marco Hirnstein

**Affiliations:** 1Department of Biological and Medical Psychology, University of Bergen, 5009 Bergen, Norway; 2NORMENT Center of Excellence, University of Bergen and Haukeland University Hospital, 5009 Bergen, Norway; 3Department of Clinical Engineering, Haukeland University Hospital, 5021 Bergen, Norway; 4Department of Radiology, Haukeland University Hospital, 5021 Bergen, Norway; 5Division of Psychiatry, Haukeland University Hospital, 5021 Bergen, Norway; 6Department of Clinical Medicine, University of Bergen, 5021 Bergen, Norway; 7Mohn Medical and Imaging Visualization Centre, Haukeland University Hospital, 5021 Bergen, Norway; 8Department of Education, UiT/The Arctic University of Norway, 9019 Tromsø, Norway; 9School of Health Sciences, Kristiania University College, 0107 Oslo, Norway

**Keywords:** schizophrenia, MR spectroscopy, brain stimulation, structural MR, resting-state functional MRI (fMRI), task-related fMRI

## Abstract

Background: Transcranial direct current stimulation (tDCS) is used as treatment for auditory verbal hallucinations (AVH). The theory behind the treatment is that tDCS *increases* activity in prefrontal cognitive control areas, which are assumed to be *hypo*active, and simultaneously *decreases* activity in temporal speech perception areas, which are assumed to be *hyper*active during AVH. We tested this hypofrontal/hypertemporal reversal theory by investigating anatomical, neurotransmitter, brain activity, and network connectivity changes over the course of tDCS treatment. Methods: A double-blind, randomized controlled trial was conducted with 21 patients receiving either sham or real tDCS treatment (2 mA) twice daily for 5 days. The anode was placed over the left dorsolateral prefrontal cortex (DLPFC) and the cathode over the left temporo-parietal cortex (TPC). Multimodal neuroimaging as well as clinical and neurocognitive functioning assessment were performed before, immediately after, and three months after treatment. Results: We found a small reduction in AVH severity in the real tDCS group, but no corresponding neuroimaging changes in either DLPFCD or TPC. Limitations: The study has a small sample size. Conclusion: The results suggest that the currently leading theory behind tDCS treatment of AVH may need to be revised, if confirmed by studies with larger N. Tentative findings point to the involvement of Broca’s area as a critical structure for tDCS treatment.

## 1. Introduction

A groundbreaking study [1] found a substantial reduction in auditory verbal hallucinations (AVH) in medication-resistant patients with schizophrenia after two daily sessions of transcranial direct current stimulation (tDCS) over the course of five days. A current of 2 mA was applied with the anode over the left dorsolateral prefrontal cortex (DLPFC) and the cathode over the left temporo-parietal cortex (TPC). Since then, inconsistent results were reported. Some studies replicated the reduction in AVH [2,3,4,5], some reported no effect of tDCS relative to sham [6,7,8], and some reported improvements in other symptoms than AVH [9,10,11,12,13]. Several reviews concluded that there is currently too little evidence *if* tDCS treatment reduces AVH, calling for more randomized controlled trials (RCT) [14,15,16,17,18,19].

Moreover, little is known about *how* the treatment works. Often, the following theory is provided: AVH arise from hyperactivity in speech perception regions in the TPC and simultaneous hypoactivity in the prefrontal cortex, which impairs top-down control over speech perception and AVH [20,21]. Prior to tDCS, transcranial magnetic stimulation (TMS) had been used to reduce AVH, mostly with TMS protocols aiming to inhibit the TPC. The results are inconsistent so far [22]. Hence, it was suggested [1] that the bipolar effects of tDCS could be exploited in order to stimulate both the TCP and DLPFC simultaneously.

By placing the typically excitatory anode over the hypoactive DLPFC area and the typically inhibitory cathode over the hyperactive TPC, it is assumed that the hypofrontality and hypertemporality is reversed and AVH frequency and severity reduced [1]. However, it is not certain whether this theory (henceforth termed “hypofrontal/hypertemporal reversal theory”) holds: Recent simulation/modelling studies have demonstrated that the peak of the electric field with a unihemispheric DLPFC/TPC montage lies between the two electrodes, in Broca’s area, rather than in DLPFC and TPC areas themselves [23,24]. Moreover, the assumption that the anode and cathode are generally excitatory and inhibitory, respectively, is no longer tenable [25].

Few tDCS studies on AVH included neuroimaging to examine the underlying neural mechanisms of the tDCS treatment. In one RCT, anodal tDCS over the DLPFC alone indicated changes in functional connectivity in frontal-thalamic-temporo-parietal networks in the subgenual cortex and DLPFC [9]. Mondino et al. [26] found that the treatment effect with anodal DLPFC and cathodal TPC stimulation correlated with resting-state connectivity between the left TPC and the left anterior insula. An EEG study found a significant increase in gamma event-related synchronization in the left DLPFC in context with significant improvement of working memory performance [27]. Despite these advances, studies so far have focused on a single neuroimaging modality and did not provide a direct test of the hypofrontal/hypertemporal reversal theory. Moreover, there is acknowledged need amongst the tDCS community to investigate the underlying neural mechanism of tDCS with multimodal neuroimaging approaches [28].

We therefore carried out a double-blind RCT with two parallel arms (real and sham tDCS) that included extensive multimodal neuroimaging allowing us to map anatomical changes with structural magnetic resonance imaging (MRI), neurochemical changes with MR spectroscopy, simulation/modelling of the tDCS electric field, as well as functional activity and connectivity with task-related and resting-state functional MRI (fMRI), respectively.

The main objective was to uncover the underlying neuronal mechanisms of tDCS treatment for AVH and to directly test the hypofrontal/hypertemporal reversal theory. A secondary objective was to provide more data on the efficacy of the tDCS treatment. We hypothesized that the tDCS treatment would (a) reduce AVH, (b) increase glutamate levels (the main excitatory neurotransmitter), and decrease GABA levels (the main inhibitory neurotransmitter) underneath the anode in the DLPFC, relative to sham stimulation. Correspondingly, we hypothesized decreased glutamate and increased GABA levels under the cathode in the TPC. We hypothesized that the fMRI measures would indicate increased activity in the DLPFC and decreased activity in the TPC, if the tDCS treatment were successful. Moreover, we expected stronger connectivity between the DLPFC and TPC. The inclusion of structural measures was exploratory. The electric field was modelled to visualize individual differences and to control whether differences in current delivery arose between the real and sham tDCS group.

## 2. Methods

The RCT was registered on clinicaltrials.gov (“Transcranial Brain Stimulation and Its Underlying Neural Mechanisms as a Novel Treatment for Auditory Hallucinations” #NCT02769507). All procedures involving human participants/patients were approved by Reginal committees for medical and health research Ethics (REK), *#*2014/2179.

The preregistered primary outcomes were Auditory Hallucination Rating Scale (AHRS), Questionnaire for Psychotic Experiences (QPE), Positive and Negative Syndrome Scale (PANSS), Hallucinations app, and Hallucinations change scale (HCS). The preregistered secondary outcomes were Stroop task, Trailmaking test A and B, Expectations Questionnaire, Adverse Effects Questionnaire, Clinical Global Impressions Scale, Global Assessment of Functioning, Shape, size, and integrity of gray and white matter structures in the brain; GABA and glutamate levels in the dorsolateral prefrontal cortex; and peri-Sylvian regions, Resting state functional MRI, Dichotic listening paradigm, BOLD response during dichotic listening paradigm, white matter structure, and connectivity. All outcomes are reported herein, besides the QPE and hallucinations app due to too little data. White matter structure and connectivity was changed to arterial spin label measure early in the RCT, but also here, too few datasets were obtained for meaningful analysis.

### 2.1. Participants

Twenty-four participants (both in and outpatients) were recruited between January 2017 to January 2020, mainly from the Division of Psychiatry, Haukeland University Hospital (Bergen, Norway) but also other psychiatric units in the Bergen area via flyers, posters, and national media. All participants had been diagnosed by their independent psychiatrist according to ICD10 [29]. Three participants dropped out because they found participation too challenging, leaving a sample of N = 21 (real tDCS: n = 11, sham: n = 10). Demographic data of the participants are presented in Table 1 and Figure 1 shows the CONSORT flow diagram (http://www.consort-statement.org/ (accessed on 14 September 2022)). Missing data are due to participants being unable to complete the clinical assessment or finding neuroimaging/dichotic listening too challenging. The full CONSORT checklist can be found on page 12–14 in the Appendix A.

Inclusion criteria were experiencing AVH episodes at least five times a week, at least two weeks of stable antipsychotic medication, unsuccessful treatment of AVH with at least two different antipsychotics, and ability to provide informed consent. Exclusion criteria were younger than 18 years of age, being under guardianship or mandatory mental health care, metal implants/splinters, severe skin disease in the area of electrode placement, and pregnancy. These criteria were adopted from Koops et al. [6] to allow pooling together the clinical data from both studies at a later stage. Koops et al. [6] did not include neuroimaging data.

#### Blinding

Before the project started, a colleague at our department, not connected to the project, was asked to make a sequence of randomly alternating sham and real tDCS codes. Those codes were provided by the tDCS manufacturer and either trigger sham or real tDCS when entered into the tDCS machine. When the first patient came, the first code on the list was used, the second patient received the second code and so forth. There were parallel code sequences for male and female participants to ensure a similar gender distribution across the real/sham condition.

Before the final session, the PI, who was not involved in testing/assessing the patients, gave the project staff (i.e., those administering tDCS and assessing the participants) a closed envelope with a note reading either “real stimulation” or “sham stimulation”. This envelope was opened by the staff together with participants after the follow-up session was completed and after both participants and project staff completed the blinding- questionnaire and guessed whether the participant had received real or sham tDCS. In case participants received sham, they were offered real stimulation, in accordance with the ethical approval of the study.

### 2.2. tDCS Treatment

tDCS was given twice daily for 20 min and for five consecutive days at 2 mA (current density = 0.057 mA/cm), with a minimum 3 h break in between. tDCS parameters were exactly modelled after Brunelin et al. [1]. Before and after each 20 min of real stimulation, there was a 30 s ramp-up and 30 s ramp-down period. In sham, after 30 s ramp-up, tDCS was delivered for 40 s, followed by 30 s ramp-down and very weak pulses of 110 µA lasting 15 ms, provided every 550 ms as an impedance check. The 5 × 7 cm electrodes were inside a sponge soaked in a 9 mg/mL NaCl solution and coated with electrode gel. Electrode positions for tDCS were located with EEG caps, based on the 10/20 system. The anode and cathode were placed over AF3 (left DLPFC) and CP5 (left TPC), respectively. We used a *neuroConn DC Stimulator Plus* (neuroCare GmbH, Ilmenau, Germany) and ‘Signa gel’ electrode gel (BIOPAC Systems Inc., Santa Barbara, CA, USA) for facilitating current flow. The electrode positions for tDCS were located with EEG caps (EASYCAP GmbH, 82211 Herrsching, Germany). The participants were not engaged in a specific behavioral task during tDCS, they were asked to sit and relax with eyes open but without any visual or audio stimuli. Conversation with the researchers was kept to a minimum but was not always avoidable depending on the participant and their needs in order to be comfortable in the stimulation situation.

## 3. Procedure and Materials

Prior to participation, potential candidates were informed about the procedures and screened for inclusion and exclusion criteria. If eligible and interested in participating, written informed consent was obtained from all patients. We collected demographic and clinical data via self-report questionnaires, interviews, and their medical records. Participants also provided their medication regime for the last two weeks.

Participants completed a clinical assessment, neuroimaging, and neurocognitive/general functioning assessment three times: at baseline (i.e., before tDCS treatment), directly after tDCS treatment (henceforth “post-treatment”), and three months after tDCS treatment (henceforth “follow-up”). Assignment to real or sham tDCS was pseudo-randomized and gender was blocked. An overview of the RCT design is provided in Table 2. Not all participants were able to complete all assessment tools at all three time points. The exact sample sizes for each assessment tool are provided in Figure 1. The study was finished as planned in 2020 due to the ending of funding period.

### 3.1. Clinical Assessment

Before treatment, participants answered a treatment expectations questionnaire, ranging from 0 (“treatment has no effect”) to 10 (“treatment has full effect and AVH will fully disappear”). Clinical assessment included the AHRS, a self-report measure on the severity of AVH comprising seven items (range: 2–41) [30,31], Apathy Evaluation Scale (AES) (18 items, range 18–72, added after pre-registration), PANSS (30 items, range 30–210) [32], and QPE [33,34]. At post-treatment, participants completed the AHCS [31], a self-report tool, where participants indicate the percentage by which their AVH improved, got worse, or stayed the same on a scale from −100 (hallucinations doubled) to +100% (hallucinations disappeared).

Due to the significant difference between real and sham groups in AHRS baseline scores (Table 1), we computed change measures for the AHRS as follows:Δ post-treatment = AHRS_Session2_ − AHRS_Session1_(1)
Δ follow-up = AHRS_Session3_ − AHRS_Session1_.(2)

Thus, positive values indicate an increase in AVH, while negative values indicate a decrease in AVH. The values were transformed into percentages and subjected to 2 × 2 repeated measures ANOVAs with the between-participants factor *tDCS treatment* (real/sham) and the within-participants factor *time* (post-treatment change/follow-up change). Measures from the other clinical assessment tools were subjected to 2 × 3 ANCOVAs with *tDCS treatment* and *time* (baseline/post-treatment/follow-up). For these measures, difference scores were not used as there were no baseline differences between groups. Baseline AHRS scores were used as a covariate in this and all subsequent analyses.

### 3.2. Neuroimaging

For MRI, a 3T GE (General Electric, Chicago, IL, SA) Discovery MR750 Scanner and a GE 32 Channel head coil (Invivo corp., Gainsville, FL, USA) were used. Each MR session took 54 min. Before entering the scanner, participants trained the dichotic listening task and underwent a screening for hearing deficits (see Appendix A for details on paradigms).

#### 3.2.1. Structural Measures

Structural anatomical images were acquired with a 3D T1 weighted fast spoiled gradient sequence (FSPGR): TE = 3 ms, TR = 6.8 ms, FOV = 256 mm, number of slices = 188, slice thickness = 1 mm (no gap) flip angle = 12 degrees, and matrix = 256 × 256. Parcellation of the cortical surface was performed using Freesurfer (version 5.3.0, Athinoula A. Martinos Center for Biomedical Imaging (2013). FreeSurfer (Software), available from https://surfer.nmr.mgh.harvard.edu/ (accessed on 14 September 2022)) with the Destrieux atlas [35]) to yield estimates for surface area, average cortical thickness, and gray matter volume (amongst other parameters not examined in the present study) for each of the 74 labelled regions per hemisphere.

Data analysis: 15 participants completed structural scans in all three sessions (real n = 9, sham n = 6). Exploratory t-tests were carried out for all 74 regions per hemisphere provided by the analysis comparing baseline and post-treatment for the whole group and the real and sham group individually. Additionally, treatment effects were investigated in depth in the following 12 brain regions, selected based on our regions of interest, the left DLPFC and TPC, subcentral gyrus and sulci, transverse frontopolar gyri and sulci, middle frontal gyrus, supramarginal gyrus, lateral aspect of the superior temporal gyrus, planum temporale, posterior ramus of the lateral sulcus, inferior frontal sulcus, middle frontal sulcus and superior temporal sulcus to explore the Broca’s area, the opercular part of the inferior frontal gyrus and triangular part of the inferior frontal gyrus. Then, 2 × 3 ANCOVAs with tDCS treatment (real/sham) and time (baseline/post-treatment/follow-up) were computed, one for each brain region and measurement (surface area, gray matter volume, and average cortical thickness).

#### 3.2.2. MR Spectroscopy

The structural scan, which was reoriented along the temporal lobe, was used to position a single MRS-voxel in the left DLPFC (27 × 25 × 32 mm, volume 21.6 mL) and in the left TPC (26 × 31 × 31 mm, volume 25.0 mL). Both areas were scanned twice, once with a standard “Point RESolved Spectroscopy” (PRESS) sequence and once with a GABA-edited MEshcher-GArwood PRESS (MEGAPRESS) sequence, resulting in four spectra. Data processing followed a well-established pipeline in our group [24,36]. PRESS data were acquired at TE = 35 ms, TR = 1500 ms, 128 repetitions plus eight water-unsuppressed reference scans acquired automatically after the metabolite scans, with 4096 samples at sampling frequency 5 kHz. Data were quantified with LCModel version 6.3-1J, using a basis set incorporating components from 15 metabolites: alanine, aspartate, creatine, γ-aminobutyric acid, glucose, glutamine, glutamate, glycreophosphorylcholine, phosphorylcholine, lactate, myo-inositol (mI), N-acetylaspartate (NAA), N-acetylaspartylglutamate, scyllo-inositol, and taurine. Due to potential limits in accurately discriminating between glutamine and glutamate at 3T, an aggregate of the two is reported herein as Glx.

GABA-edited spectra were acquired with a MEGAPRESS sequence (TE = 68 ms, TR = 1500 ms, with 16 ms sinc-weighted gaussian editing at 1.9 (ON) and 7.5 ppm (OFF), 200 repetitions of edit ON/OFF pairs followed by 16 unsuppressed water reference scans, 4096 samples at 5 kHz). Data were processed and quantified using Gannet 3.0 with the combined GABA/Glx fit algorithm, having a Gaussian model for GABA around 3.02 ppm and a dual Gaussian for Glx around 3.71 and 3.79 ppm, with downweighting in the choline region (around 3.2 ppm). Whether the DLPFC or TPC was measured first was alternated across participants and within their three MR sessions. Metabolite estimates from both methods were scaled to an internal water reference and then adjusted for partial volume effects, water concentration, and expected relaxation times in different tissue classes, according to the formula proposed in Gasparovic et al. [37]. Tissue content for each voxel was estimated from the structural T1 images after segmentation into distinct tissue classes (gray matter, white matter, and cerebro-spinal fluid) using the combined segmentation and spatial normalization algorithm in SPM12 (https://www.fil.ion.ucl.ac.uk/spm/ (accessed on 14 September 2022)) with individual voxel masks.

Spectral (and fit) quality was ensured by visual inspection with attention to linewidth, signal-to-noise ratio (SNR), and CRLB of individual estimates, as well as aberrations in baseline or residual signals. An example of voxel placement and a successfully acquired spectrum can be seen in Appendix A.

Two 2 × 3 × 2 ANCOVAs with *tDCS treatment* (real/sham), *Time* (baseline/ post-treatment/follow-up), and *Brain area* (DLPFC/TPC) were computed, one each for Glx (which is the sum of glutamate and glutamine, and here used as a proxy for glutamate) and GABA.

#### 3.2.3. Resting State fMRI

Resting state fMRI was performed with closed eyes (number of volumes = 160, TE = 30 ms, TR = 2000 ms, FOV = 220 mm, Slice thickness = 3 mm, slice spacing = 0.5 mm, and a 96 × 96 matrix). Data preprocessing was the same as for the task-related fMRI data.

Seed-based functional connectivity analyses in addition to a Fractional Amplitude of Low-Frequency Fluctuations (fALFF) analysis were conducted. In the Conn Toolbox (version v.17.f http://www.nitrc.org/projects/conn (accessed on 14 September 2022)), data went through a default denoising procedure, where motion realignment parameters (including their first derivatives) and time courses from cerebrospinal fluid (CSF) and white matter (WM) were regressed out (first five components from a Principal Component decomposition for a CSF mask as well as a WM mask).

Seed-based functional connectivity identifies regions where activity is correlated with the activity in a seed region, by computing a cross-correlation between the BOLD signal time-series of the seed and the rest of the brain [38].

fALFF is a measure showing relative BOLD signal power within the frequency band of interest (0.008–0.09 Hz in this study) compared to the entire frequency band and is defined as a ratio of root mean square of BOLD signal at each individual voxel after vs. before low- or band-pass filtering [39]. fALFF is regarded as an indicator of spontaneous neural activity [39,40] since it coincides with other established activity measures [41] and shows the expected Default Mode Network activity patterns during rest [39,42].

Two seed regions of interest were generated from the prescribed MRS voxel masks in the left DLPFC and TPC (see Appendix A). For each of the individual MRS voxels, volumetric masks were generated in participant-space to describe the prescribed coverage of that voxel. The structural image on which those voxels were prescribed was transformed into standard space, first by a linear affine transformation and then by non-linear warping. Voxel masks for each individual participant were then overlaid, and the 80-percentile of coverage taken point wise to identify those regions covered by the respective voxels in 80% of participants. tDCS treatment groups were compared in different contrasts. For seed-to-voxel connectivity and fALFF, a cluster correction procedure at the single-voxel level was applied with thresholds of *p* < 0.001 and *p* < 0.005, respectively. For both, the cluster level threshold was *p* < 0.05 and multiple comparisons were FDR-corrected.

#### 3.2.4. fMRI/Dichotic Listening Paradigm

Participants (real tDCS: n = 6, sham: n = 5) carried out the well-established fMRI-adaptation of the Bergen dichotic listening paradigm [43,44], in which participants have to discriminate between syllables that are presented concurrently to each ear. The number of correct responses (in percentage) was determined from verbal responses during scanning.

The screening for hearing deficits ensured that all participants could detect frequencies between 250 Hz and 3000 Hz at an intensity of less than 20 dB deviation from a norm sample. In the Bergen dichotic listening paradigm [43,44], two out of six different syllables (/ba/, /da/, /ga/, /pa/, /ta/, and /ka/) are presented simultaneously in each dichotic listening trial, one syllable to the left and one syllable to the right ear. Since homonyms (e.g., /pa/-/pa/) are not included, there are 30 possible syllable combinations. Participants were instructed to verbally report the syllable they heard best and most clearly (no attentional focus), or they were asked to specifically report the stimulus from the left and right ear (attentional focus: left and right, respectively). Stimuli were presented with E-Prime 2.0 Professional (Psychology Software Tools Inc., Sharpsburs, PA, USA).

The dichotic listening paradigm was carried out in a block design during fMRI acquisition, using a 2D gradient echo-planar imaging sequence with the following parameters: TE = 30 ms, nominal TR = 3500 ms (1500 ms nominal acquisition + 2000 ms “silent gap”, flip angle = 90 degrees, 64 × 64 matrix, FOV = 220 mm, ~27 axial slices of 5 mm thickness with 0.5 mm gap. The paradigm had 140 volumes, distributed across 13 blocks (4 resting-blocks + 3 blocks no attentional focus + 3 blocks attentional focus right + 3 blocks attentional focus left). Each block had 10 trials, resulting in 90 dichotic listening trials and 40 resting volumes. After each volume a silent gap was provided for presenting the stimuli and recording the verbal response [45].

This task was chosen because it has previously been shown to produce reliable activation in the DLPFC and TPC areas [45]. Moreover, healthy individuals typically display behaviorally a right ear advantage, indicative of left-hemispheric language lateralization that is reduced in patients with schizophrenia [46], especially in those with frequent and severe AVH [47]. Schizophrenia patients are also less able to attend to the stimulus presented in one particular ear [48]. We thus hypothesized that real tDCS relative to sham tDCS would lead to a behaviorally larger right ear advantage and reduce the attentional impairment.

The fMRI data were preprocessed using SPM12 by realigning and unwarping the data to correct for movement and related image distortions, normalization into the MNI standard reference space, and smoothing with an 8 mm Gaussian kernel to improve SNR. The data were visually inspected for remaining artefacts. For subsequent analyses, the first four dummy scans were removed. First level analysis was performed for each participant and session by specifying a general linear model that incorporated the onsets of the stimulation blocks of the three conditions and included the realignment parameter as regressors of no interest. Contrasts were defined for exploring the effect of each of the three attentional focus conditions (none, right, left) separately. The high-pass filter was set to 364 Hz.

Accuracy rates were subjected to a 2 × 3 × 2 × 3 ANCOVA with *tDCS treatment* (real/sham), *Time* (baseline/post-treatment/follow-up), *Ear* (left/right), and *Attentional focus* (none/left/right). The fMRI data were analyzed with a group analysis for individual contrast images in a 2 × 3 ANCOVA with *tDCS treatment* (real/sham) and *Time* (baseline/post-treatment/follow-up).

#### 3.2.5. Simulation of Electrical Field

The simulation of the tDCS electrical field in each participant was performed based on the corresponding individual structural scans (n = 15) obtained at baseline (n = 12) or, if not applicable, the post-treatment (n = 1) or follow-up (n = 2) structural scans were used. For the simulation of the electrical field, SimNIBS 2.1.2 (Simulation of NIBS (Non-Invasive Brain Stimulation (Version 2.1.2) (Software) available from www.simnibs.org) was used [49,50]. A Mesh model was created of each participant’s head in FreeSurfer. To run the simulation model, the electrodes in the simulation were placed on the 10/20 EEG cap system provided by SimNIBS; the anode over AF3 and the cathode over CP5 to simulate the real tDCS setup as closely as possible. The simulated electrodes were identical to the ones used for actual tDCS (5 × 7 cm^2^, 1 mm electrode thickness, 8 mm sponge thickness).

Means for the 99% percentile of the electric field strength and for focality (the gray matter volume with an electric field greater or equal to 75% of the peak values) were computed and compared between real and sham tDCS groups with Mann–Whitney U tests.

#### 3.2.6. Correlations

To investigate how changes in the brain were related to changes in AVH over the course of the treatment, we computed Pearson correlations between the change-scores in AHRS (described above) and change-scores for Glx, GABA, structural measures, and the electric field in the real tDCS group. Pearson correlations were used as a more sensitive measure, although not all variables were normally distributed. Change-scores for Glx/GABA/structural measures were computed with the same formula as for AHRS change scores:Δ post-treatment = Glx/GABA/structural_measure_Session2_ − Glx/GABA/structural_measure_Session1_(3)
Δ follow-up = Glx/GABA/structural_measure_Session3_ − Glx/GABA/structural_measure_Session1_(4)

### 3.3. Neurocognitive/General Functioning

Neurocognitive/general functioning assessment entailed: Clinical Global Impression (CGI) [51]; Global Assessment of Functioning (GAF) [52]; Stroop test, Trail Making Test [53]; and the Norwegian version [54] of the National Adult Reading Test (NART), a measure for premorbid IQ [55].

Outcome measures were subjected to 2 × 3 ANCOVAs, one for each measure, with *tDCS treatment* (real/sham) and *Time* (baseline/post-treatment/follow-up).

## 4. Results

Effect sizes are provided as partial eta-squared (η_p_^2^) or Cohen’s *d*.

### 4.1. Clinical and Neurocognitive/General Functioning Assessment

AHRS change scores showed no significant *tDCS treatment*, *time*, or interaction effect (all *Fs_(1,18)_* ≤ 0.90, *ps* ≥ 0.355, η_p_^2^s ≤ 0.048). However, the intercept was significant (*F* = 20.00, *p* < 0.001, η_p_^2^ = 0.526), indicating there was a significant reduction of AVH by 15% (*SD* = 3%) across both treatment groups and both timepoints. More specifically, there was a 12% reduction in AVH with real tDCS and 15% with sham, post-treatment. This difference corresponds to *d* = 0.14. At follow-up, the sham group had a 12% reduction in AVH, the real tDCS group a 21% reduction compared to baseline (Figure 2). The difference was *d* = −0.47. Negative *d* values denote larger AVH improvement in real vs. sham groups.

Based on the AHCS (n = 20), 5/11 (45.5%) of the participants in the real group and 5/9 (56%) of the sham group reported that their AVH scores improved (χ² = 0.202, *p* = 0.653). The average improvement with real tDCS (*M* = 24.9%, *SD* = 34%, range 0−99%) was not significantly different from sham improvement (*M* = 21.7%, *SD* = 28%, range 0−85%), (*F*_18_ = 0.42, *p* = 0.524). Only participants with real tDCS differed significantly from zero (*t*_10_ = 2.40, *p* = 0.038), but there was also a trend in the sham group (*t*_8_ = 2.28, *p* = 0.052), according to one-sample t-tests. Notably, none of the patients reported worse AVH after treatment.

None of the other clinical or neurocognitive/general functioning measures showed any significant effects. The descriptive means and ANCOVA results are displayed in Table 3.

### 4.2. Structural Analysis

There was no significant main effect or interaction involving *tDCS treatment* in any of the three anatomical measures in any of the 12 brain regions of interest—even before correction for multiple comparisons. Additionally, the exploratory analysis of 74 regions per hemisphere did not yield significant effects. The results for all 74 regions per hemisphere can be found in the Appendix A, page 2–4.

### 4.3. MR Spectroscopy

There were no significant *tDCS treatment*, *time* effects, or interactions for Glx (all *Fs_(1,13)_* ≤ 1.71, *ps* ≥ 0.203, η_p_^2^s ≤ 0.124) or GABA (all *Fs_(1,13)_* ≤ 2.89, *ps* ≥ 0.115, η_p_^2^s ≤ 0.194); see Appendix A. Means are provided in the Appendix A, page 5.

### 4.4. Resting State fMRI

Seed to voxel connectivity: With the TPC as seed across all participants (i.e., independent of whether participants received real or sham tDCS), we found a negative connectivity cluster in the left and right superior frontal gyrus when comparing post-treatment with baseline (cluster peak coordinates xyz = [−10 50 34 mm], cluster size k = 486 voxels, *p* = 0.00006, Figure 3A). Post hoc analyses per *time* showed that this connectivity cluster was not present during baseline but appeared as negative connectivity at post-treatment.

*f*ALFF: When comparing real and sham tDCS between post-treatment and baseline, we found a cluster in the precentral gyrus (peak xyz = [−56 8 23], k = 117, cluster *p_size_*_p-FDR_ = 0.047, Figure 3B). Post hoc t-tests for values extracted from this cluster showed that the means for the real tDCS group were significantly (*t*(8) = 2.92, *p* = 0.019) lower at post-treatment (*M* = 0.333 *SD* = 0.60) as compared to baseline (*M*=0.908 *SD*=0.31). In sham, there was no significant difference (*t*(5) = 0.76, *p* = 0.482) between baseline (*M* = 0.463 *SD* = 0.49) and at post-treatment (*M* = 0.707 *SD* = 0.72) (Figure 3C).

Functional connectivity between the TPC and the DLPFC did not change significantly between baseline and treatment time points, as indicated by an analysis of correlations between the two ROIs’ time courses. There were no significant findings in the comparisons between baseline and follow-up in connectivity or brain activity.

### 4.5. fMRI/Dichotic Listening Paradigm

There were no significant findings regarding tDCS for the task-related fMRI data. A t-contrast across all conditions showed that the DL task resulted in the typical activation of auditory cortices (*p* < 0.05, FWE), Appendix A.

The behavioral data did not reveal any significant main or interactions effects involving the factor *tDCS treatment* (all *Fs_(2,16)_* ≤ 3.42, *ps* ≥ 0.058, η_p_^2^s ≤ 0.300). None of the remaining main or interactions effects were significant (all *Fs _(2,16)_* ≤ 3.42, *ps* ≥ 0.058, η_p_^2^s ≤ 0.300).

### 4.6. Simulation of Electrical Field

For the whole cortex, the 99% percentile peak field was *M* = 0.37 (*SD* = 0.06 V/m) and focality was *M* = 8131 (*SD* = 1684 mm^2^). These simulation parameters did not differ significantly between real and sham groups (all *U*s ≤ 16, *ps* ≥ 0.195). Simulations for each participant are provided in the Appendix A, page 8.

### 4.7. Correlations

When corrected for multiple testing with Holm’s method [56], none of the correlations reached significance for changes in AHRS scores against (a) changes in Glx and GABA (all *rs* ≤ −0.71, *ps* ≥ 0.034, *ps_corr_* ≥ 0.54), (b) changes in structural measurements for all 12 selected brain areas (all *rs* ≤ 0.72, *ps* = 0.030, *ps_corr_* =1.0), and (c) electric field strength and focality values (all *rs* ≤ −0.57, *ps* ≥ 0.113, *ps_corr_* ≥ 0.90). Correlations between MRI measures are reported in the Appendix A, page 9.

## 5. Discussion

The present study tested the underlying neural mechanisms of the most common tDCS treatment for AVH with a multimodal neuroimaging approach. In short, we found a small clinical improvement of AVH post-treatment and at follow-up but only sparse effects of tDCS on various neuroimaging parameters.

***Auditory verbal hallucination assessment***. Both self-reporting tools (AHCS and AHRS) showed a small reduction of AVH that was similar in real tDCS and sham groups. The AHCS (i.e., participants’ ratings of how much AVH changed) revealed a 25% reduction in AVH with real and 22% with sham tDCS. The AHRS, which covers more aspects of AVH than the AHCS questionnaire, yielded roughly 15% improvement and, again, there was a slightly stronger improvement for real versus sham tDCS at follow-up (*d* = −0.47).

No improvements were found when the experimenter rated “hallucinatory behavior” (PANSS-P3 item), negative symptoms, positive symptoms, and the total PANSS score. To conclude, tDCS had a small, positive effect on AVH alone that was slightly above a placebo effect (sham) but difficult to identify by others than the participants themselves.

The positive effect on AVH was smaller than in some previous studies [1,5] but of similar magnitude as in Koops et al. [6], who also did not find improvements beyond placebo. This discrepancy could be explained by the fact that both Brunelin, Mondino, Gassab, Haesebaert, Gaha, Suaud-Chagny, Saoud, Mechri and Poulet [1] and Lindenmayer, Kulsa, Sultana, Kaur, Yang, Ljuri, Parker and Khan [5] tested samples with schizophrenia patients only, while we and Koops, Blom, Bouachmir, Slot, Neggers and Sommer [6] had a mixed sample. However, most of our participants either had schizophrenia (71%) or some type of psychosis.

***Neuroimaging***. On a **structural** level, several studies reported that AVH in schizophrenia were associated with gray matter reductions in several areas including the insula, right superior temporal and fusiform gyri, left inferior and superior temporal gyri (comprising Heschl’s gyrus) (see reviews [57,58,59]). On a **functional fMRI and behavioral** level, we found the well-established expected activation in the auditory cortex (see Appendix A, page 7) [60] and the expected absence of a right ear advantage in schizophrenia patients in dichotic listening [47]. On a **neurotransmitter** level, schizophrenia patients, in general, show reduced Glx levels in the temporal and frontal lobe compared to healthy individuals [61]. However, the tDCS treatment had no significant effect on structural, functional fMRI, behavioral dichotic listening, Glx, and GABA measures. To our knowledge, this was the first time such effects had been studied. The lack of neurotransmitter findings, however, are in line with two previous studies, in which we failed to detect Glx changes after tDCS over the DLPFC and TPC in healthy individuals [24,62].

During **resting state fMRI,** with a less conservative single-voxel threshold of 0.005, brain activity (as measured with fALFF) in the right precentral gyrus significantly decreased from baseline to post-treatment in the real tDCS group, while it remained unchanged in the sham group. Note that this decrease would not become significant with a more conservative 0.001 single-voxel threshold, but we felt it important to report this trend for future investigations. Furthermore, the connectivity between the TPC and right/left superior frontal gyrus became significantly negative post-treatment while there was no significant connectivity at baseline. However, since it emerged in both real and sham tDCS, it is not in line with the hypofrontal/hypertemporal reversal theory.

In the literature, AVH has been associated with connectivity reductions in the uncinate fasciculus, corpus callosum, thalamus radiation, and fronto-occipital fibers [58] and the primary and secondary auditory cortex [59]. Previous studies correlated the AVH reduction after tDCS with altered connectivity between the DLPFC with a frontal-thalamic-temporo-parietal network [9] and between the TPC with various regions including the left DLPFC [26]. The latter findings are particularly relevant as the authors used the same montage as in the present study and found a correlation between the altered connectivity and AVH reduction. A possible explanation for the inconsistent findings may be that the treatment effect was much larger in Mondino et al. [26], possibly due to a more homogeneous sample of schizophrenia patients and slightly different inclusion criteria than in the present study.

***General discussion of multimodal neuroimaging.*** In summary, neither brain activation, nor brain structure, nor Glx/GABA levels showed significant effects of tDCS in the DLPFC or TPC, and none of those parameters correlated with changes in AVH over the course of the treatment. Thus, our data are not in line with the leading theory behind the AVH treatment with tDCS. Furthermore, the simulation of tDCS showed that the main effect of the electric current was not underneath the electrodes, as predicted by the hypofrontal/hypertemporal reversal theory, but between the electrodes, largely in Broca’s area, consistent with results from healthy individuals [24] and other patients [23]. Although we investigated several neuroimaging parameters, it is well possible that none of them picked up significant changes due to the small sample size. It is also possible that the hypofrontal/hypertemporal reversal pattern only emerges in treatment responders. For example, a recent study found that tDCS responders had a higher electric field in the left transverse temporal gyrus than tDCS non-responders [23]. We refrained from an analysis of treatment responders due to the small sample size.

## 6. Limitations

As pointed out above, a limitation of the present study is the small sample size. The initial goal was to recruit 60 participants to double the sample size of the original Brunelin et al. study [1]. This seemed realistic within the four year funding period, based on the experience in our research group with the same patient population in previous studies. However, participants often hesitated to commit to the study’s extensive assessment regime (6 full days and follow-up), in addition to general skepticism towards brain stimulation. Moreover, a change in Norway’s health policy meant that potential candidates were released much earlier from our major collaboration clinic to ambulant treatment facilities. This made recruitment more difficult.

A sensitivity analysis for a 2 × 3 ANCOVA with G*Power [63] revealed an effect size of *f* ≥ 0.73 with the parameters: *n* = 21, α = 0.05, power = 0.80, numerator df = 2, and number of groups = 2. Given that *f* = 0.4 is widely considered a large effect [64], we only had enough power to detect large and very large effects with certainty.

Given the low N, a more homogenous sample in terms of diagnose might have been preferable. We included participants without schizophrenia to pool our clinical data together with another study [6] and to increase the sample size. Real tDCS participants appear more heavily medicated than sham participants, which might have reduced the AVH improvement. Given the small sample, it was not possible to correct for this difference statistically. However, higher medication is in line with a higher AHRS score (reflecting more severe AVH), which was used as a covariate.

## 7. Conclusions

As null findings in small samples are inherently difficult to interpret, our study obviously does not refute the leading theory behind tDCS treatment of AHV. However, given that none of the various neuroimaging measures showed even a trend, our study certainly calls the hypofrontal/hypertemporal reversal theory into question. There is mounting evidence that Broca’s area plays a much more crucial role than previously thought. Multi-center collaborations are needed to increase sample size and to provide more reliable conclusions regarding the efficacy of the tDCS treatment for hallucinations.

## Figures and Tables

**Figure 1 brainsci-12-01382-f001:**
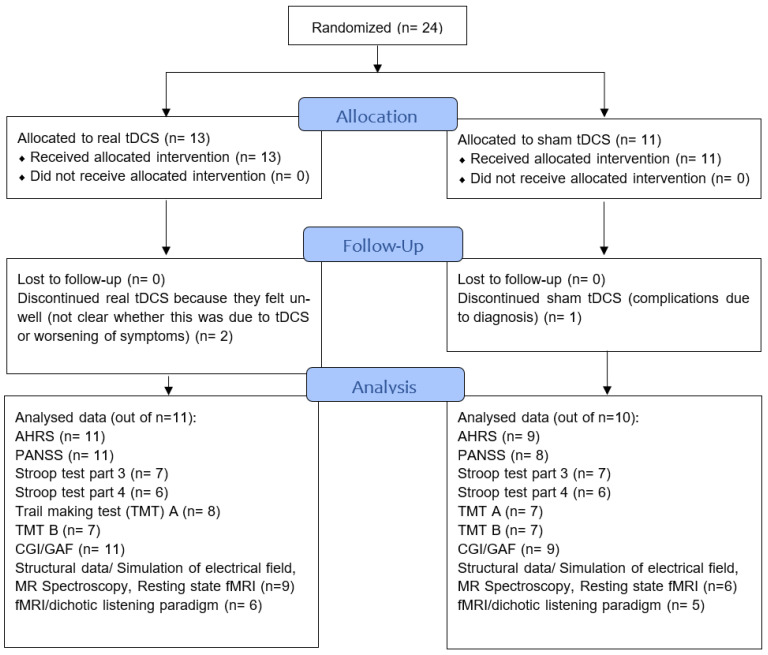
CONSORT flowchart of the study showing sample sizes throughout study, follow-up and analysis. AHRS—Auditory Hallucination Rating Scale, PANSS—Positive and negative syndrome scale, CGI—Clinical Global Impression. GAF—Global Assessment of Functioning, Stroop—Stroop test, TMT—Trail Making Test, fMRI—functional magnetic resonance imaging.

**Figure 2 brainsci-12-01382-f002:**
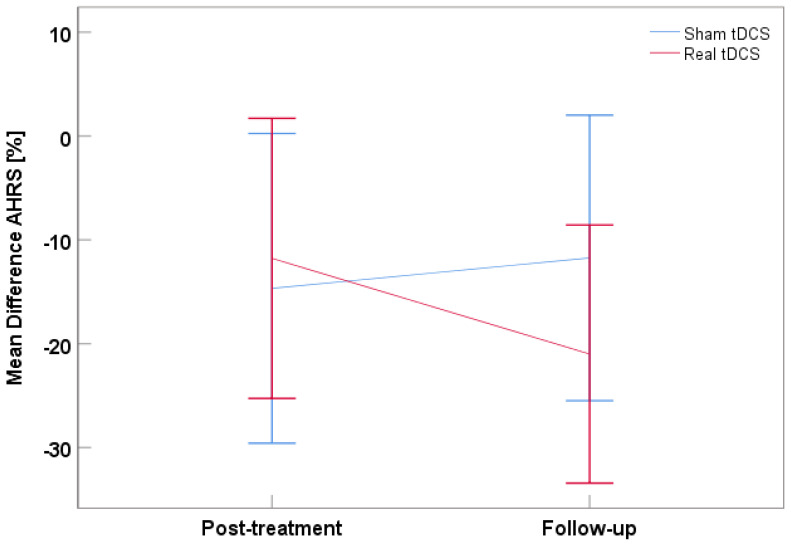
Mean difference for the AHRS score at post-treatment and follow-up with the real tDCS group in red and the sham tDCS groups in blue. Negative values represent a reduction, positive values an increase in AVH relative to baseline. Error bars denote 95% CI.

**Figure 3 brainsci-12-01382-f003:**
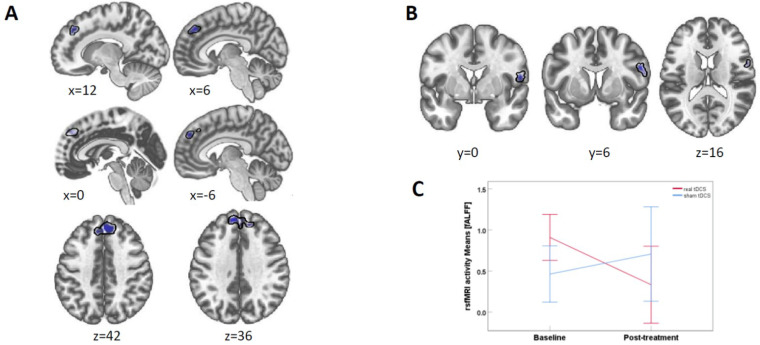
Resting-state fMRI results. (**A**) TPC-seeded connectivity with the superior frontal gyrus cluster (shown in blue) across all participants (i.e., real and sham group combined) was reduced for post-treatment compared to baseline. p-threshold at 0.001. (**B**) fALFF brain activity in a right precentral gyrus cluster between real and sham for baseline vs. treatment at p-threshold 0.005. (**C**) fALFF activity means for the right precentral gyrus cluster. The change in the tDCS group is significant (*p* = 0.019), while it is not in the sham group. Error bars denote 95% CI.

**Table 1 brainsci-12-01382-t001:** Sample demographics at baseline.

	tDCS	Sham tDCS	Statistics
	n = 11	n = 10	
Age in years (range, M(SD))	22–68, 38 (13)	19–47, 33 (10)	*t*_19_ = 0.927, n.s.
Gender (male/female)	8/3	6/4	χ² = 0.382, n.s.
Handedness (right/left)	10/1	8/2	χ² = 0.509, n.s.
Education in years (range, M(SD))	8–18, 12 (3)	9–15, 11 (2)	*t*_19_ = 0.496, n.s
Defined Daily Dosage (M(SD)) ^1^	1.86 (0.98)	0.99 (1.3)	*t*_19_ = 1.727, n.s.
Duration of illness (in years M(SD))	10 (5)	14 (10)	*t*_18_ = 0.953, n.s.
Treatment expectation	6.8	6	*t*_19_ = 0.766, n.s.
Nicotine users (total n)	8	8	χ² = 0.153, n.s.
Blinding patient (correct/incorrect)	8/3	5/5	χ² = 1.147, n.s.
Blinding researcher (correct/incorrect)	5/6	7/3	χ²=1.289, n.s.
PANSS score total sum M(SD)	76 (25)	69 (14)	*t*_19_ = 0.770, n.s.
AHRS score M(SD)	**30(5)**	**24(4)**	***t*19 = 2.83, *p* = 0.010**
NART (total errors, M(SD))	24 (14)	20 (7)	*t*_14_ = 0.619, n.s.
Diagnosis ^2^			
Paranoid Schizophrenia	8	5	
Other schizophrenia types		2	
Unspecific nonorganic psychosis	2		
Schizoaffective disorder, depressive type	1		
Paranoid personality disorder		1	
Persistent delusional disorders		1	
Recurrent Depressive disorder		1	
Antipsychotic medication ^2^			
First-generation	1		
Second-generation	7	6	
Both	3		
None		4	

n.s.—not significant, PANSS—Positive and negative syndrome scale, AHRS—Auditory hallucinations rating scale, NART—National Adult Reading Test. For descriptions of all parameters see chapters “Clinical assessment” and “Neurocognitive/general functioning”. ^1^ of antipsychotics. ^2^ No statistical tests were calculated because n was too low.

**Table 2 brainsci-12-01382-t002:** Overview of the procedure and timeline of the RCT is shown.

		Randomized Control Trial			
	Pre-Study Assessment	Day 1 Baseline	Day 2 to 5	Day 6 Post-Treatment	3 Months Follow-Up
tDCS treatment			2× daily 20 min of 2 mA tDCS	2× 20 min 2 mA tDCS	
Clinical examination	tDCS check-list, screening questionnaire, medication	Hearing test	Adverse Effects Questionnaire	Adverse Effects Questionnaire	
Hallucination assessment		AHRS, QPE, PANSS		AHRS, QPE, PANSS, AHCS	AHRS, QPE, PANSS
Neuroimaging		Session 1: structural MRI, MR spectroscopy, Resting state fMRI, dichotic listening fMRI, ASL.		Session 2: structural MRI, MR spectroscopy, Resting state fMRI, dichotic listening fMRI, ASL.	Session 3: structural MRI, MR spectroscopy, Resting state fMRI, dichotic listening fMRI, ASL.
General functioning and neurocognitive abilities	Pre-study Questionnaire, Informed consent	AES, NART, CGI, GAF, Stroop, TMT, General Information Questionnaire, Expectations Questionnaire		AES, CGI, GAF, Stroop, TMT	AES, CGI, GAF, Stroop, TMT, Blinding Check Questionnaire

AHRS—Auditory Hallucination Rating Scale, AES—Apathy Evaluation Scale, PANSS—Positive and negative syndrome scale, QPE—Questionnaire of psychotic experiences, CGI—Clinical Global Impression. GAF—Global Assessment of Functioning, Stroop—Stroop test, TMT—Trail Making Test, NART—National Adult Reading Test, AHCS—auditory hallucinations change scale.

**Table 3 brainsci-12-01382-t003:** Mean scores for all clinical and neurocognitive/general functioning tools for real and sham participants at all time points.

Test (Score Range/Unit)	Baseline		Post-Treatment		Follow-Up		rmANCOVA Results
	Real	Sham	Real	Sham	Real	Sham	
PANSS negative sum (7–49)	22	16	22	15	22	17	all *F_(2,32)_s* ≤ 2.37, *ps* ≥ 0.110, η_p_^2^s ≤ 0.129
PANSS positive sum (7–49)	18	19	18	15	18	18	all *F_(2,32)_s* ≤ 1.42, *ps* ≥ 0.257, η_p_^2^s ≤ 0.081
PANSS P3 (1–7)	5.0	4.8	4.6	4.3	4.7	4.5	all *F_(2,32)_s* ≤ 0.48, *ps* ≥ 0.624, η_p_^2^s ≤ 0.029
PANSS general sum (30–210)	36	33	35	34	35	33	all *F_(2,32)_s* ≤ 0.80, *ps* ≥ 0.461, η_p_^2^s ≤ 0.047
AHRS total sum (2–41)	30	24	27	21	24	22	
AES total sum (18–72)	48	46	52	41	52	48	all *F_(2,34)_s* ≤ 1.82, *ps* ≥ 0.178, η_p_^2^s ≤ 0.096
CGI (0–7)	4.91	4.56	5.00	4.44	4.91	4.44	all *F_(2,34)_s* ≤ 0.26, *ps* ≥ 0.774, η_p_^2^s ≤ 0.015
GAF-S (0–100)	43	46	43	45	43	44	all *F_(2,34)_s* ≤ 0.35, *ps* ≥ 0.706, η_p_^2^s ≤ 0.020
GAF-F (0–100)	45	48	45	48	44	49	all *F_(2,34)_s* ≤ 0.65, *ps* ≥ 0.531, η_p_^2^s ≤ 0.037
Stroop 3 (in s)	73	77	66	63	80	69	all *F_(2,22)_s* ≤ 0.60, *ps* ≥ 0.556, η_p_^2^s ≤ 0.052
Stroop 3 mistakes (nr. Of errors)	1.43	1.14	1.43	1.86	1.14	1.57	all *F_(2,22)_s* ≤ 1.05, *ps* ≥ 0.368, η_p_^2^s ≤ 0.087
Stroop 4 (in s)	81	81	67	74	81	75	all *F_(2,18)_s* ≤ 0.99, *ps* ≥ 0.392, η_p_^2^s ≤ 0.099
Stroop 4 mistakes (nr. Of errors)	2.33	2.43	2.00	2.29	4.00	1.86	all *F_(2,20)_s* ≤ 1.44, *ps* ≥ 0.261, η_p_^2^s ≤ 0.126
TMT A (in s)	36	34	37	31	34	29	all *F_(2,24)_s* ≤ 1.46, *ps* ≥ 0.253, η_p_^2^s ≤ 0.108
TMT B (in s)	133	93	110	82	154	86	all *F_(2,22)_s* ≤ 0.15, *ps* ≥ 0.865, η_p_^2^s ≤ 0.013

PANSS—Positive and negative syndrome scale, AHRS—Auditory hallucinations rating scale, AES—Apathy Evaluation Scale, CGI—Clinical Global Impression, GAF—Global Assessment of Functioning, TMT—trail making task.

## Data Availability

Anonymized data will be made available to colleagues upon request to the corresponding author.

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
