# Peer review of "Pilot-RCT Finds No Evidence for Modulation of Neuronal Networks of Auditory Hallucinations by Transcranial Direct Current Stimulation"

_brainsci, 2022, doi:10.3390/brainsci12101382_

Round 1

Reviewer 1 Report (Previous Reviewer 2)

Thank you for giving me the opportunity to review the revised manuscript.

I think this manuscript would be suitable for publication in this journal.

Author Response

Thank you for the review and a great cooperation! 

Kind regards, Lynn Marquardt

Reviewer 2 Report (Previous Reviewer 1)

Marquardt and colleagues have extensively reviewed the manuscript entitled “Pilot-RCT finds no evidence for modulation of neuronal networks of auditory hallucinations by transcranial direct current stimulation”. The authors amply provided responses to my doubts and questions. Even though the results remain limited by the number of subjects included, I appreciate the clarification they made to the manuscript, highlighting it as the main limitation and providing more information about blinding and structural and functional analysis. The manuscript has been extensively edited and its readability has improved. 

Author Response

Thank you for the review and a great cooperation! 

Kind regards, Lynn Marquardt

This manuscript is a resubmission of an earlier submission. The following is a list of the peer review reports and author responses from that submission.

Round 1

Reviewer 1 Report

The study by Marquardt et al. entitled “Pilot-RCT finds no evidence for modulation of neuronal networks of auditory hallucinations by transcranial direct current stimulation” provides an exploratory attempt to clarify the effects of tDCS over frontal and temporal regions through a multimodal neuroimaging approach in patients with auditory verbal hallucinations (AVH).

The scope of this study is severely limited by the number of subjects included, which does not allow for a full understanding (and a clear interpretation) of the results. Because the authors declare that the literature on the field is very heterogenous, I do not think that their results help in clarify if the treatment could be useful in reducing AVH. Also, the multimodal neuroimaging approach used is not helpful, since the results are presented as separate analyses and not integrated into a model. The subjects included in the neuroimaging analyses are very low to draw any conclusion and do not provide enough information about how the treatment works. The best way to understand it should be to perform a concurrent tDCS-fMRI study. 

MAJOR REVISION

METHODS

Figure 1 is useful to understand how many patients underwent any condition and measurement, however, I think the neuroimaging part is not clear. If in the bracket are reported the patient that underwent any task, why did specify the patients that considered the ‘neuroimaging too challenging’?

Where the pharmacological treatment used similar within the patients? Did you check for any correlation between the pharmacological treatment and the outcome of tDCS treatment?

Where the protocol double-blinded?

More specifications about the ATLAS used for the structural image parcellation are needed. Moreover, what is the reason why you performed an analysis of the structural data? The rationale behind this analysis, even if exploratory, should be clarified.

Why was the resting state fMRI performed with eyes closed? How did you check the patients were awake and focused? Were the patients aware of not falling asleep and not focusing their thought on any particular tasks?

In paragraph 3.2.3 you stated that ‘Two seed regions of interest were generated from the MRS voxel masks in the left DLPFC and TPC’. However it is not clear to me how these seeds were created, and if they were masked with Grey matter. Could you please provide more specifications?

How many simulations of the tDCS were obtained using the post-treatment or follow-up structural scans instead that the baseline structural scan? It would be interesting to include this information.

Which structural measures were used for the correlations? How these structural measures were extracted? Please clarify these aspects

RESULTS

In the resting-state fMRI results, all t values are missing. Please include them together with the p values.

In the supplementary material, figure S6, you stated that ‘no systematic differences between the two groups are observed.’ Did you perform any statistical analysis? I think you can not declare this, at least if you don’t have performed statistical analysis on the e-field extracted from the ROI of interest for the two groups.

In general, I found very hard to follow the flow in the main manuscript, and even if many information/analyses are reported in the supplementary material, there are not enough references for them in the manuscript.

DISCUSSION

Generally, the discussion seems more like a repetition of the results section. Because the authors used a multimodal neuroimaging approach to investigate their hypothesis, it would be more appropriate to interpret and discuss the results in a multimodal way, thus considering the results altogether and not discussing them separately.

MINOR REVISION

Check the puntuaction (pag 2, line 56)

Please rephrase this sentence: ‘The fMRI measures were hypothesized to show increased activity 86 in the DLPFC and decreased activity in the TPC, with strengthened connectivity between 87 these areas should the tDCS treatment be effective.’ (page 2, line 86-88).

On page 6, line 193 you declared that ‘Data preprocessing was the same as for the task-related fMRI data’. However, the preprocessing steps were described in the supplementary material and after the description of resting-state fMRI. I think it would be better to move this information to the Resting-state fmri paragraph before describing the analysis (fALFF etc..).

Reviewer 2 Report

Thank you for giving me the opportunity to review this manuscript.

I think it is necessary to revise this manuscript.

1)  I think that the introduction section is well written and so interesting, but it is still necessary to explain more. The authors hypothesized that "the tDCS treatment would (a) reduce AVH, (b) increase glutamate levels (the main excitatory neurotransmitter) and decrease GABA levels (the main inhibitory neurotransmitter) underneath the anode in the DLPFC, relative to sham stimulation." However, the authors says that "the assumption that the anode and cathode are generally excitatory and inhibitory, respectively, is no longer tenable." On the other hand, high-frequency rTMS and low-frequency rTMS are generally excitatory and inhibitory, and the polarity-dependent effects of rTMS are easier to obtain than those of tDCS. If possible, please explain the results of previous rTMS studies about the above-mentioned hypothesis?? Why was tDCS selected in this study (rather than high/low frequency rTMS)??

2) Please attach the CONSORT checklist and fill in the tables.

3) Please describe any rationale for numbers in the pilot trial (initially, sample size had been estimated to be 60??). Furthermore, please explain how sample size is estimated to detect statistically significant differences in future studies. That is because the authors explained in the discussion "Although we investigated several neuroimaging parameters, it is well possible that none of them picked up significant changes due to the small sample size." Please explain why sample size was smaller than previous study [1].

4) Please explain any method used to generate the random allocation sequence.

5) Please explain any mechanism used to implement the random allocation sequence (such as sequentially numbered containers), and describe any steps taken to conceal the sequence until interventions were assigned

6) Please describe who generated the random allocation sequence, who enrolled participants, and who assigned participants to interventions.

7) Please explain who were blinded after assignment to interventions (for example, participants, tDCS administrators, those assessing outcomes) and how they were blinded.

8) Please define the one time point of the primary outcome.

9) In the discussion section, the authors explained "A possible explanation for the inconsistent findings may be that the treatment effect was much larger in Mondino et al. [25] than in the present study.", but it is also necessary to explain why the treatment effect was much larger in Mondino et al [25] than in the present study. Was the tDCS protocol better?? Was the study design different?? 

10) I agree with the authors' point that "Participants often hesitated to commit to the study’s extensive assessment regime, in addition to general  skepticism towards brain stimulation." Please explain how to improve recruitment strategies in future studies. Please re-define the optimized criteria of participants in future studies.

I think that it is necessary to revise the manuscript.
